# Manipulability Optimization of a Rehabilitative Collaborative Robotic System

**Giorgia Chiriatti** [1,*] , **Alessandro Bottiglione** [2] **and Giacomo Palmieri** [1]

1. Department of Industrial Engineering and Matemathical Scieces, Polytechnic University of Marche, 60121 Ancona, Italy; g.palmieri@univpm.it
2. Studio Pacinotti S.r.l., 60131 Ancona, Italy; s109622@studenti.univpm.it
* Correspondence: g.chiriatti@univpm.it

**Abstract:** The use of collaborative robots (or cobots) in rehabilitation therapies is aimed at assisting and shortening the patient's recovery after neurological injuries. Cobots are inherently safe when interacting with humans and can be programmed in different working modalities based on the patient's needs and the level of the injury. This study presents a design optimization of a robotic system for upper limb rehabilitation based on the manipulability ellipsoid method. The human–robot system is modeled as a closed kinematic chain in which the human hand grasps a handle attached to the robot's end effector. The manipulability ellipsoids are determined for both the human and the robotic arm and compared by calculating an index that quantifies the alignment of the principal axes. The optimal position of the robot base with respect to the patient is identified by a first global optimization and by a further local refinement, seeking the best alignment of the manipulability ellipsoids in a series of points uniformly distributed within the shared workspace.

**Keywords:** rehabilitation robotics; collaborative robotics; design optimization; manipulability ellipsoids





## 1. Introduction

Collaborative robots (or cobots) are a new category of robots able perform tasks in cooperation with humans, simply by sharing a workspace or with real physical interaction. They can support operators in manual activities, such as manufacturing or assembly tasks, in total safety thanks to advanced sensor systems, limited power and forces and ergonomic features that protect against mechanical and electrical risks [1]. This novel philosophy of robotics has evolved as one of the key drivers of Industry 4.0. Human–Robot Interaction (HRI), in particular, is a promising strategy for achieving higher and more flexible productivity by combining the decision-making ability of humans with the repeatability of robots [2]. In addition to force sensors used to determine the contact forces with the environment, cobots typically also exploit vision systems able to perceive the presence and location of objects or humans in the workspace, increasing flexibility and real time adaptability to dynamically varying scenarios [3].

Because collaborative robots are inherently safe and reliable, scientific research on their use in the healthcare sector is growing. In robotic rehabilitation, for example, the robot is in contact with patients and aims to provide physical interaction driven by the actuation systems [4], so the use of cobots may represent an appropriate choice. Currently, there is a wide range of robotic devices used in neuro-muscolar rehabilitation, starting with exoskeletons, which are rigid anthropomorphic structures directly attached to human's body segments, or end-effector devices which are usually attached to a distal segment of the patient [5]. Cobots can be considered to belong to the second category; however, only one cobot specifically designed for rehabilitation is nowadays in the market. This is ROBERT, by Life Science Robotics, which can be used for the rehabilitation of lower limbs or mobilization of legs of bedridden patients [6]. Compared to the traditional therapy, a cobot-assisted therapy can provide intensive and task-specific solutions for each patient. Moreover, it is possible to control the interaction force with the patient and, at the same

time, to record data of the motion resulting from the exercise. A further advantage is given by the possibility of carrying out long and repeated intensive therapy sessions with limited intervention by the therapist. The latter has the role of selecting the correct rehabilitation treatment among the pre-programmed exercises, supervising several patients simultaneously. At the same time, patients can train more independently and maximize their efforts [7].

In a rehabilitative cobotic system the patient's limb is typically fixed to the robot's end-effector, and the robotic manipulator is used to drive the patient arm over a path or to give a force feedback to the patient while executing a task. In general, different working modalities are possible [8–10]:

- Passive mode—the patient's limb is passive and driven by the cobot along a predefined trajectory;
- Active mode—the patient actively performs the exercise while the robot can exert a programmable resistance;
- Active-assisted mode—the patients tries to execute the task while the robot provides assistance only if the patient exhibits a lack of strength.

In general, cobot-assisted therapy is more efficient if actively assisted exercises are performed, as brain stimuli are more intense than in passive mode [11]. In order to increase the potentialities of the exercise, the authors have conceived a specific working mode, that can be named vision-assisted mode, which exploits a smart camera integrated to the robotic system used to detect a real object placed by the therapist within the manipulator workspace; when the patient is asked to reach the target object, the robot reacts with a force feedback in order to channel the movement of the hand along the correct path, possibly with active assistance to facilitate the motion in that direction. The combination of different types of feedback as visual, auditory and haptic, proves to be highly beneficial since it maximizes the attention to the task and enhances the motor performance [12].

Although the evidence on the efficacy of robot-assisted therapy is growing, there are still problems related to the lack of standardized protocols and the differences between the various devices that can be used [13]. However, the advantages of cobot rehabilitation, such as repeatability, high intensity and limited intervention by therapists, are the prerequisites for a rapid spread of this practice compared to traditional therapies.

This paper presents a design optimization of a robotic system for upper limb rehabilitation based on the manipulability ellipsoid method. An optimization algorithm is used to find the best location of the robot's base with respect to the human shoulder in order to confer to the human and robotic arms a similar kinematic behaviour when the are coupled. The problem, typical of other application fields such as machining operations [14], can be approached by different methods [15,16]. In this case, the human–robot system is modeled as a closed kinematic chain in which the human hand grasps a handle attached to the robot's end-effector. The manipulability ellipsoids are determined for both the human and the robotic arm and compared by calculating an index that quantifies the alignment of the principal axes. The optimal position of the robot base is identified by a first global optimization on a predefined grid of points and by a further local refinement, seeking the best alignment of the manipulability ellipsoids in a series of points uniformly distributed within the shared workspace.

Section 2 describes the kinematic model of the human and robotic arms. Section 3 introduces the velocity and force ellipsoids used to define the index which describes the alignment of principal directions of manipulability between the two kinematic chains. The design optimization procedure is presented in Section 4, where the main results are also discussed, while conclusions and future works are given in Section 5.

## 2. Kinematic Model of the Human–Robot System

As shown in Figure 1a, the human–robot system consists of a closed kinematic chain in which the human hand grasps a handle fixed to the end effector of the commercial cobot Universal Robots UR5e, already used by the authors in a series of studies in the field of

human–robot collaboration [17]. The ergonomic handle, suitably made for a comfortable grip, is shown in Figure 1b.

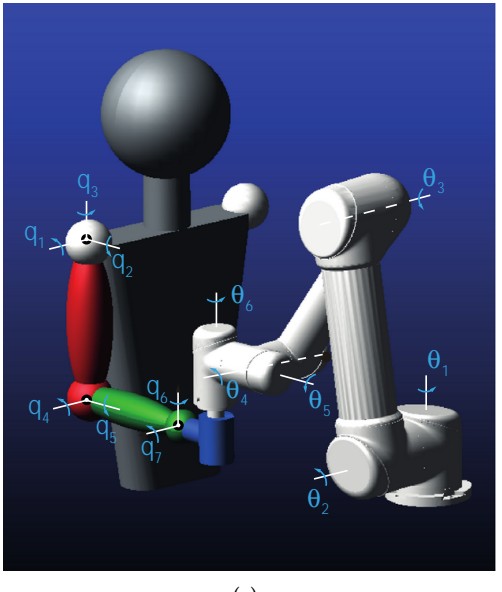 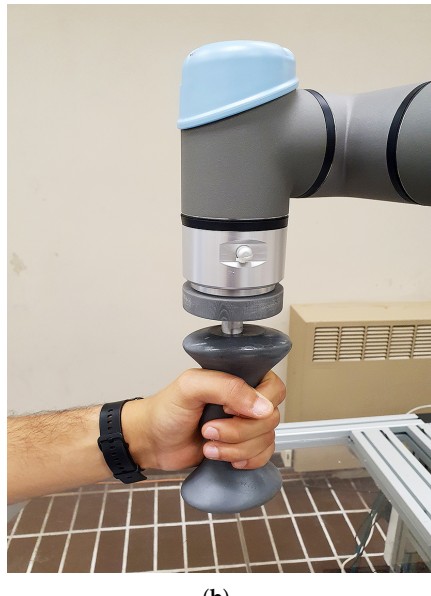

(a)                                                         (b)

**Figure 1.** (**a**) Kinematic chain of the human–robot system; (**b**) ergonomic handle attached to the end-effector.

The human arm is modeled as three rigid segments connected by frictionless joints with seven degrees of freedom (DOF) in total. The spherical joint representing the shoulder confers flexion–extension ($q_1$), abduction–adduction ($q_2$) and internal–external rotation ($q_3$). The elbow is modeled as a universal joint that allows for flexion–extension ($q_4$) and pronation–supination ($q_5$) of the forearm. The universal joint relative to the wrist provides the flexion–extension ($q_6$) and the ulnar–radial deviation of the hand ($q_7$). To confine joint rotations within physiological limits, the maximum and minimum angles are set according to the values available from the OpenSim software [18] (Table 1). The Italian male 50th percentile is considered as a reference for anthropometric measurements. Table 2 summarizes the lengths of the body segments; the length of the hand, closed to hold the handle, is considered half of the total for simplicity.

The robot UR5e is characterized by a serial chain of revolute joints arranged as shown in Figure 1a which confers a full mobility (6 DOF) to the end-effector; joint angles are hereafter indicated as $\theta_i$ with $i = 1, \ldots, 6$.

The kinematics of the human and robotic arms are implemented by the Matlab Robotic Toolbox using the Denavit–Hartemberg (DH) method, the parameters of which are summarized in Tables 3 and 4. The resulting kinematic chains are represented in Figure 2a,b for the human and robotic arms, respectively.

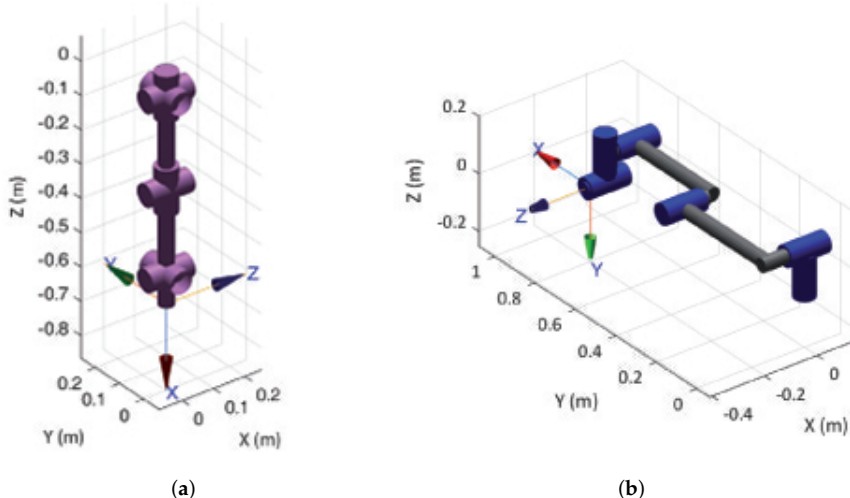

|  |  |
|:---:|:---:|
| (**a**) | (**b**) |

**Figure 2.** Kinematic models of the human arm (**a**) and the UR5e (**b**) implemented in Matlab Robotic Toolbox.

**Table 1.** Joint limits of the human arm.

| Joint | Minimum Value [°] | Maximum Value [°] |
|:---:|:---:|:---:|
| $q_1$ | −90 | 90 |
| $q_2$ | −120 | 90 |
| $q_3$ | −90 | 90 |
| $q_4$ | 0 | 150 |
| $q_5$ | 0 | 180 |
| $q_6$ | −70 | 70 |
| $q_7$ | −25 | 35 |

**Table 2.** Lengths of the upper limb segments (50th percentile Italian male).

| Upper Limb Segment | Length [mm] |
|:---:|:---:|
| Height | 1750 |
| Arm | 280 |
| Forearm | 256 |
| Hand | 189 |
| Closed Hand | 95 |

**Table 3.** DH parameters of the human arm: d is the distance along x-axis of the current joint; a is the distance along x-axis between two consecutive joint axis; $\alpha$ is the rotation around the x-axis of the current joint; off-set is the angle between the two consecutive x-axis about the z-axis of the previous joint [19].

| Joint | d [m] | a [m] | $\alpha$[°] | Off-Set [°] |
|:---:|:---:|:---:|:---:|:---:|
| $q_1$ | 0 | 0 | 90 | 90 |
| $q_2$ | 0 | 0 | 90 | 90 |
| $q_3$ | −0.27 | 0 | 90 | 90 |
| $q_4$ | 0 | 0 | −90 | 0 |
| $q_5$ | −0.25 | 0 | 90 | 0 |
| $q_6$ | 0 | 0 | 90 | 90 |
| $q_7$ | 0 | −0.09 | −90 | 0 |

**Table 4.** DH parameters of the UR5e robot (see definitions given in Table 3).

| Joint | d [m] | a [m] | $\alpha$ [°] | Off-Set [°] |
|---|---|---|---|---|
| $\theta_1$ | 0.09 | 0 | 90 | 90 |
| $\theta_2$ | 0.14 | −0.42 | 0 | 0 |
| $\theta_3$ | −0.12 | −0.39 | 0 | 0 |
| $\theta_4$ | 0.11 | 0 | 90 | 0 |
| $\theta_5$ | 0.09 | 0 | 90 | 180 |
| $\theta_6$ | 0.05 | 0 | 0 | 0 |

The inverse kinematics of the human arm is solved by a numerical approach that aims to minimize the error function $e(\mathbf{q}) = |\mathbf{f}(\mathbf{q}) - \mathbf{x}|$ starting from a guess solution $\mathbf{q_0}$, being $\mathbf{f}(\mathbf{q})$ the direct kinematics law and $\mathbf{x} = [x, y, z, \alpha, \beta, \gamma]^T$ the Cartesian pose of the hand. The $x, y, z$ sequence of current rotation axes corresponding to the rotation angles $\alpha, \beta, \gamma$ is used to represent the orientation. Furthermore, the minimization procedure is implemented taking into account physiological limits of joint rotations.

The velocity kinematics of the human arm can be formulated as:

$$\dot{\mathbf{x}} = \begin{bmatrix} \dot{\mathbf{x}}_l \\ \omega \end{bmatrix} = \begin{bmatrix} \mathbf{J}_p \\ \mathbf{J}_o \end{bmatrix} \dot{\mathbf{q}} = \mathbf{J}(\mathbf{q}) \dot{\mathbf{q}} \tag{1}$$

where the velocity vector $\dot{\mathbf{x}}$ is composed by the linear velocity vector $\dot{\mathbf{x}}_l$ and the angular velocity $\omega$, while $\mathbf{J}(\mathbf{q})$ is the geometrical Jacobian matrix of dimension $(6 \times 7)$, composed by $\mathbf{J}_p$ and $\mathbf{J}_o$ which are the $(3 \times 7)$ position and orientation Jacobian matrices, respectively. A similar approach for both position and velocity kinematics is used for the robotic arm, which is constrained to realize the same motion of the human hand in the Cartesian space acting on the six DOF related to actuated joints $\theta_i$ with $i = 1, \ldots, 6$. Figure 3 shows the human and robot models in the rest position of the human arm ($\mathbf{q} = [50° \, 0 \, 0 \, 33° \, 90° \, 0 \, -6°]^T$), corresponding to the robot joint position vector $\boldsymbol{\theta} = [-261° \, 207° \, -47° \, -70° \, 90° \, 9°]^T$. Without loss of generality, the origin of the global coordinate system is located on the shoulder of the human joint.

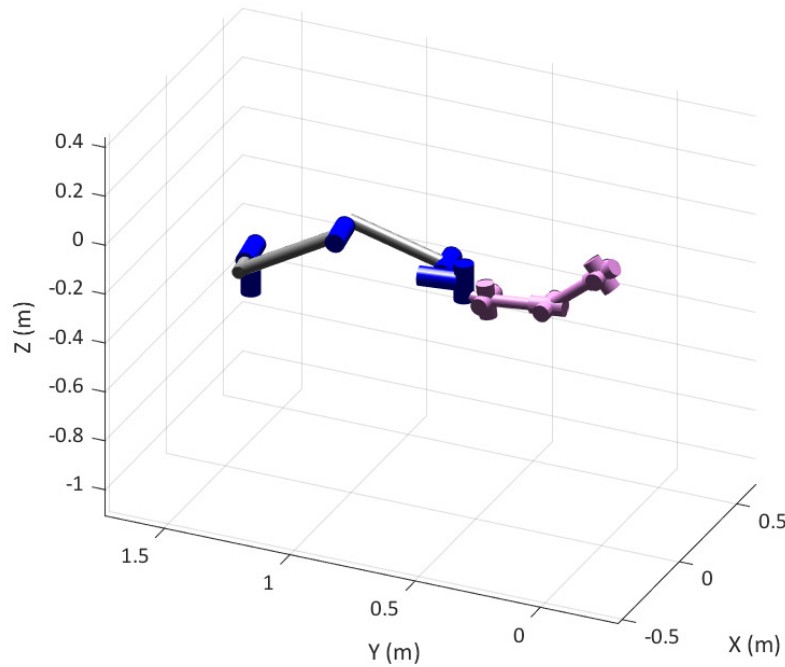

**Figure 3.** Closed kinematic chain of human and robotic arm models.

A set of points of the shared human–robot workspace is defined in order to evaluate the average kineto-static affinity of the two arms in a uniform spatial distribution. Using

spherical coordinates with the center coincident with the human shoulder, two radii are considered based on the total length of the upper limb. They correspond to the 83% and 50% of the total upper limb length, respectively. Abduction/adduction of the shoulder is spanned by $\pm 30°$, whereas the flexion/extension range is $\pm 20°$. A total of 18 points are in this way defined, as shown in Figure 4. The orientation of the hand on each of the points is defined by a local frame which has always the x axis aligned to the forearm and the z axis aligned with the vertical direction.

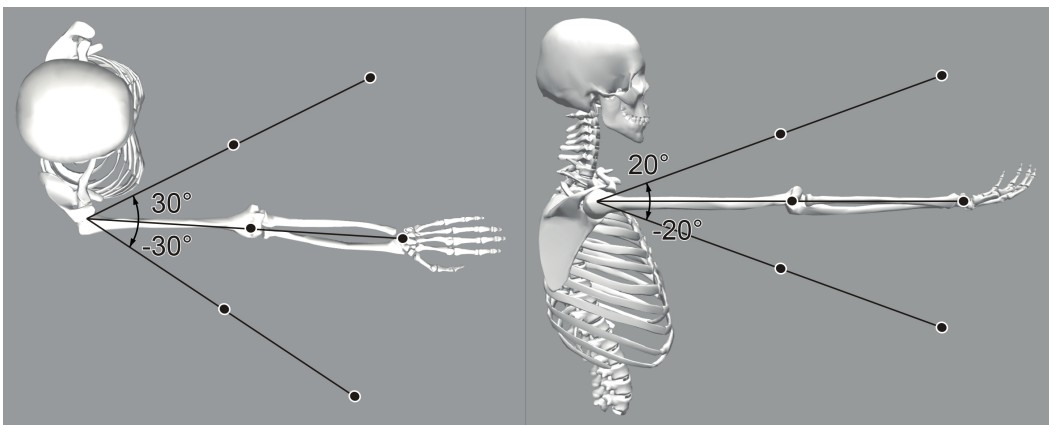

**Figure 4.** Set of points used for the evaluation of the objective function of the optimization.

### 3. Manipulability Analysis

Several studies are available in the literature on manipulability analysis on human and robotic arms. An index based on the intersection volume of velocity ellipsoids is used in [20], where the human arm (modeled with 5 DOF) and a KUKA collaborative robot are considered. In [21], a robotic-assistive control system for the rehabilitation of the human arm is studied analyzing the principal axes of the manipulability ellipsoids in order to find the easiest direction of motion of the upper limb. Other studies, as [22], focus on the relationship between the manipulability ellipsoids and the activation of the musculoskeletal system.

In general, manipulability can be defined as the capacity of change in position and orientation of the end-effector of a robot given a joint configuration [19,23]. In particular, the velocity manipulability ellipsoid describes the operational space velocities that can be generated by a given set of joint velocities with unitary norm in a known pose of the manipulator. In terms of equations, the unitary norm constraint of the joint space velocity $\dot{\mathbf{q}}$ can be expressed as:

$$\dot{\mathbf{q}}^T \dot{\mathbf{q}} = 1 \tag{2}$$

The Jacobian matrix $\mathbf{J}(\mathbf{q})$ of the manipulator can be used to map Equation (2) into the Cartesian space:

$$\dot{\mathbf{x}}^T \left( \mathbf{J}(\mathbf{q})\mathbf{J}(\mathbf{q})^T \right)^\dagger \dot{\mathbf{x}} = 1 \tag{3}$$

where † indicates the pseudo-inverse operator that must be applied in case of non-square Jacobians. As a result, the unitary radius sphere surface represented by Equation (2) transforms in an ellipsoid surface expressed by Equation (3).

Limiting the problem to translations, only the $\mathbf{J}_p$ Jacobian relative to the linear velocity of the end-effector is considered. Thus, the axes directions $\mathbf{u}_i$ of the velocity ellipsoid can be found as eigenvectors of the matrix $\left( \mathbf{J}_p(\mathbf{q})\mathbf{J}_p(\mathbf{q})^T \right)^\dagger$, whereas their dimension $\sigma_i$ is equal to the square root of the relative eigenvalues $\lambda_i$:

$$\left( \mathbf{J}_p(\mathbf{q})\mathbf{J}_p(\mathbf{q})^T \right)^\dagger \mathbf{u}_i = \lambda_i \mathbf{u}_i \qquad \sigma_i = \sqrt{\lambda_i} \qquad i = 1, 2, 3 \tag{4}$$

The directions and dimensions of the axes of the ellipsoid describe the motion capacity of the end effector: along the major axis the end-effector can move at the maximum velocity, whereas the minor axis corresponds to the direction of minimum velocity.

According to the kinetostatic duality [24], the force ellipsoid can be obtained by calculating the eigenvectors and the eigenvalues of the matrix $\mathbf{J}_p(\mathbf{q})\mathbf{J}_p(\mathbf{q})^T$. As a result, the directions of the velocity and force ellipsoids axes are the same, whereas their dimensions are reciprocal; consequently, the two ellipsoids are orthogonal to each other. Figure 5 shows the velocity (yellow) and force (green) ellipsoids for the UR5e robotic arm in a specific configuration; as expected, the direction of maximum velocity corresponds to a minimum of force.

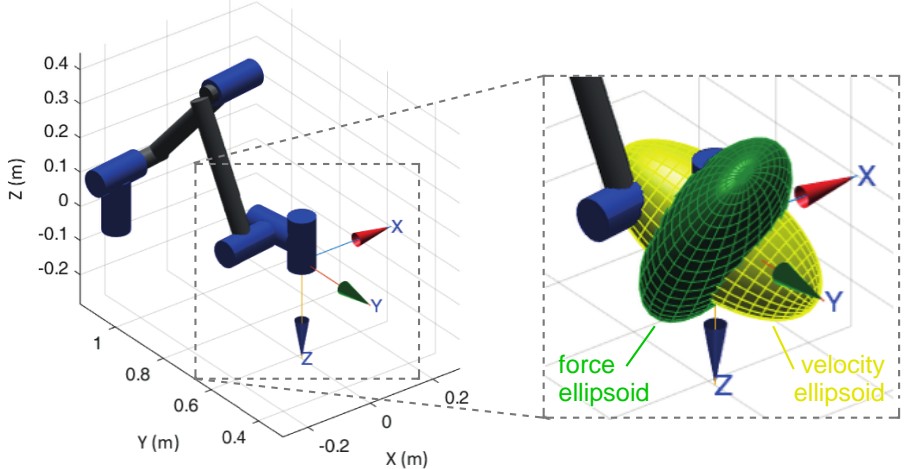

**Figure 5.** Example of ellipsoids of manipulability for the robot UR5e.

In order to evaluate the kinematic affinity between the robot and the human arm, only the velocity ellipsoids are considered in this study. Obviously, the optimal configuration of the system obtained by a kinematic (velocity) approach will correspond also to the optimal configuration from a static (force) point of view. Once joint positions of the two arms are assigned and the Jacobian matrices are calculated, ellipsoids of manipulability can be determined in the operational space and the dimensions of their axes can be normalized setting to one the maximum axis and scaling proportionally the others. As an example, Figure 6 shows the velocity ellipsoid for the robotic (a) and human arm (b) in a common pose of the end-effector, with a frame representing the axes orientation.

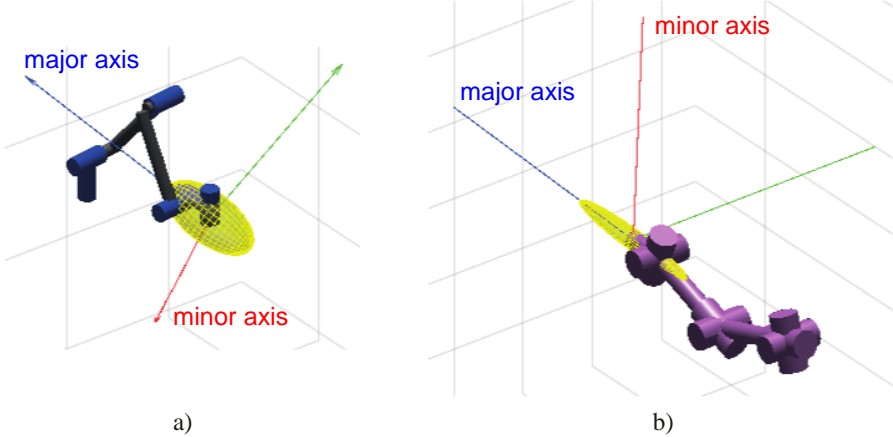

a)          b)

**Figure 6.** Velocity ellipsoids with axes orientation: (**a**) robotic arm, (**b**) human arm.

It is assumed that an optimal configuration of the system is obtained when the human and the robot have a similar ability to develop velocities along a certain direction, that is,

the ellipsoids have a similar orientation of their axes. To quantify the kinematic affinity of the two arms a scalar index can be defined as:

$$I = \frac{\sum_{i=1}^{3} |\mathbf{a}_{i,r} \cdot \mathbf{a}_{i,h}|}{\sum_{i=1}^{3} a_{i,r} a_{i,h}} \tag{5}$$

where $\mathbf{a}_i = \mathbf{u}_i \sigma_i$ is the vector representing the $i$th axis, index $i = 1, 2, 3$ indicates the order of the axis $\mathbf{a}_i$, from major ($i = 1$) to minor ($i = 3$), and subscripts $r, h$ relate to robot and human, respectively. The output is an absolute value between 0 and 1, where 0 indicates that there is orthogonality between the two ellipsoids, whereas 1 indicates a perfect alignment of them. Furthermore, the alignment of the major axis weights more than the remaining axes, especially when the ellipsoid is stretched along a principal direction. In Figure 6, for example, human and robot present almost aligned major axes, with an index value $I = 0.7$. The same index can be calculated at all the poses of the set represented in Figure 4 to evaluate the average value $I_{av}$:

$$I_{av} = \frac{\sum_{j=1}^{18} I_j}{18} \tag{6}$$

where $I_j$ is the index $I$ evaluated for the $j^{th}$ pose of the end-effector inside the workspace. The index $I_{av}$ indicates how valid the specific layout of the system is. The relative position of the base of the robot with respect to the shoulder of the man, in particular, is the free element of the problem to be obtained through an optimization procedure.

## 4. Layout Optimization

An optimization algorithm based on the evaluation of the $I_{av}$ index is implemented to find the optimal position of the robot's base with respect to the human shoulder. The optimal position is sought in a domain consisting of two horizontal planes (Figure 7), the first located at the shoulder, the second at the elbow (when the arm is extended downwards along the trunk).

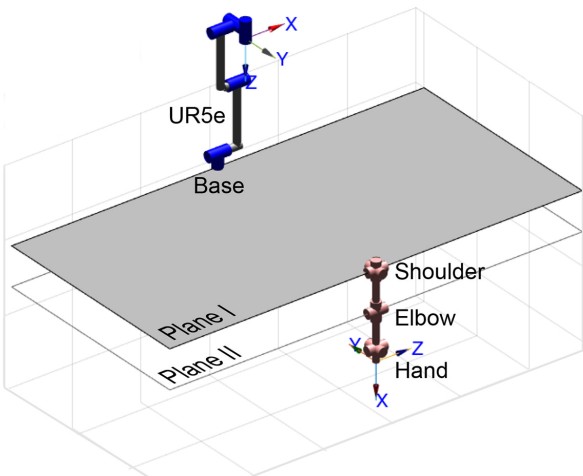

**Figure 7.** Domain of the robot's base position for the optimization algorithm.

The first step of the algorithm is the evaluation of the average index $I_{av}$ in a discrete grid of points where the robot's base is thought to be fixed. The grid is defined on planes I and II with a resolution of 100 mm. Once the base position with the highest value of $I_{av}$ is found by the initial global optimization, the output is used as guess solution for the second step of the algorithm, which is a local optimization performed by the *fminsearch* routine by Matlab; the objective function is still the average index $I_{av}$ while the optimization algorithm is based on the Nelder–Mead method (also known as downhill simplex method) which is a

numerical method used to find the minimum or maximum of an objective function in an unconstrained multidimensional space by a direct search based on function comparison.

The outputs of the global optimization algorithm are summarized in Table 5, whereas the interpolated maps of $I_{av}$ on the Planes I and II are plotted in Figure 8.

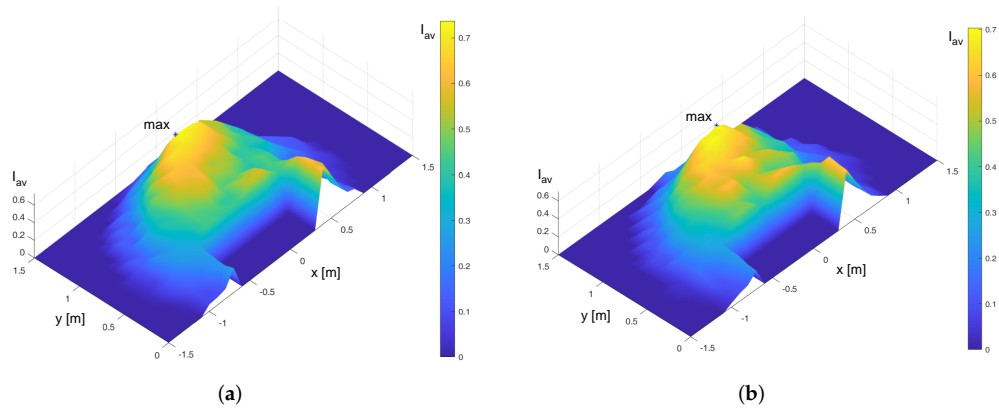

(**a**)                                                (**b**)

**Figure 8.** Interpolated maps of $I_{av}$ on plane I (**a**) and plane II (**b**).

The results obtained after the second step of local optimization are summarized in Table 6. The refined values of the optimal position of the robot base are very close to the global optimization outputs. Furthermore, a strong influence on the coordinates $x$ and $y$ can be noticed, while a variation of the height $z$ implies a small modification of the value of $I_{av}$. This result suggests positioning the robot base at $x \simeq 0$ and $y \simeq 1.1\,\text{m}$, while, for design simplicity, the base can be fixed on the desk top which is approximately at the elbow level ($z \simeq -0.3\,\text{m}$) without significantly impairing system performance. Figure 9 shows the final layout of the system prototype which is currently under experimentation.

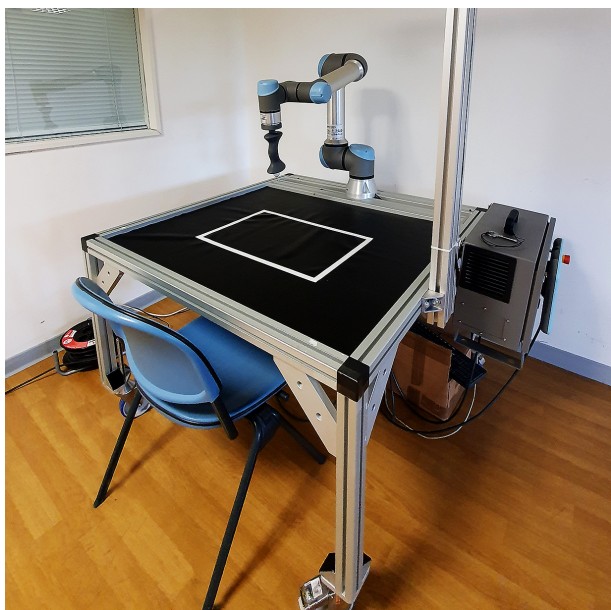

**Figure 9.** Final layout of the rehabilitation station.

**Table 5.** Output of the global optimization.

| | Base Position | | | |
|---|---|---|---|---|
| | $x$ [m] | $y$ [m] | $z$ [m] | $I_{av}$ |
| Plane I | −0.2 | 1.1 | 0 | 0.74 |
| Plane II | 0 | 1.1 | −0.3 | 0.70 |

**Table 6.** Output of the local optimization.

| | Base Position | | | |
|---|---|---|---|---|
| | $x$ [m] | $y$ [m] | $z$ [m] | $I_{av}$ |
| Plane I | −0.118 | 1.157 | 0.001 | 0.75 |
| Plane II | 0.001 | 1.158 | −0.113 | 0.74 |

**5. Conclusions**

In this work, the optimization of the layout of a collaborative robotic system for upper limb rehabilitation was presented. The optimization method was based on a manipulability analysis that quantifies the kinematic affinity between the robotic arm and the human one by means of the $I_{av}$ index that derives from the comparison of the velocity ellipsoids of the two arms. The aim was to create a system in which no constraint of velocity/force of the machine limits the ability to carry out rehabilitation exercises of various kinds.

A two-step algorithm was used to find the optimal position for the robot's base relative to the human shoulder. This result was taken into account in the final design of the system. Even if the result of the optimization procedure depends on the anthropometric parameters of the patient, a general indication can be deduced: the robot should be placed in front of the patient ($x \simeq 0$) at a distance of approximately 1 m, whereas the height of the base can range from the shoulder (plane I) to the elbow (plane II) of the patient without significant differences. Thus, the simplest solution for the design can be adopted, i.e. to collocate the robot directly on the desk top.

The system was realized and tested at the Mechatronics and Industrial Robotics Laboratory (MIR Lab) of the Polytechnic University of Marche, Ancona, Italy, where various protocols were developed, including the use of a vision system to identify a real target to be physically grasped by the patient with the active assistance of the robot. During the exercise it is possible to acquire a series of data, among which the most important are the actual trajectory of the end-effector and the force applied by the patient's hand, which can be used to define quantitative indices for monitoring the path of recovery of the patient. The first clinical trials are currently underway at the Neurorehabilitation Clinic, Azienda Ospedali Riuniti, Ancona. The preliminary results and impressions are positive and promising. The patients had good acceptance of the rehabilitation system while the therapists were able to set up the robot and supervise the therapy very easily. Future work will focus on an in-depth analysis of the data acquired during the tests, aimed at improving the control of the robot and exercise protocols in order to better meet the needs of patients and therapists that emerged from the first clinical trials.

**Author Contributions:** Conceptualization , G.P. and G.C.; investigation, A.B.; writing—original draft preparation, G.C. and A.B.; writing—review and editing, G.P.; supervision, G.P. All authors have read and agreed to the published version of the manuscript.

**Funding:** This research received no external funding.

**Institutional Review Board Statement:** Not applicable.

**Informed Consent Statement:** Not applicable.

**Data Availability Statement:** Not applicable.

**Conflicts of Interest:** The authors declare no conflict of interest.

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
