# Peer review of "Manipulability Optimization of a Rehabilitative Collaborative Robotic System"

_machines, doi:10.3390/machines10060452_

Round 1

Reviewer 1 Report

In this work, authors presented the optimal layout of a collaborative robotic system for upper limb rehabilitation. Following are my comments:

1) More literature must be added in terms of optimization methods used for placement of a robot.

2) Can you please label what is min and max in the figure 6.

3) What does value of Iav (equ 6) signifies?

4) The main contribution of this paper is to find the optimal layout of the robot. Please give more information about optimization algorithms: type of method, constraints, optimization function etc.

Reviewer 2 Report

The authors propose a solution of optimization of manipulability of a collaborative robotic system that should serve for rehabilitation. 

There are some formal issues that must be corrected. 

Page 3, line 94 - abbreviation DOF: Maybe that many readers know that DOF means "degree of freedom". However, I would recommend to add the explanation.

Page 4, tables 3 and 4: there are variables "d" and "a" - I did not find in any figure or in the text meaning of these variables. Please add it.

The authors should describe how they plan to assess the rehabilitation process. Do they plan to perfrom any objective measurement?

Do the authors already have any feedback from a rehabilitation clinic?

English needs revision. There are mostly mistyping errors.

Reviewer 3 Report

The paper topic is surely well defined, focusing on technical implementation. However, improvements can be made in the introduction (I suggest to create a background sub-section with additional references) before describing the rationale. Envisioning further applications and, possibly, ties with other methodologies (as in Digital Twins) could be interesting for the reader as well. Overall, the paper is quite complete, targeting its scope (if this matches the goals of the journal, even without an empirical demonstration with subjects, this can be published almost as it is, after a minor language check).

Round 2

Reviewer 1 Report

I would like to thank the authors for clearly addressing my comments.